# A new protocol for single-cell RNA-seq reveals stochastic gene expression during lag phase in budding yeast

Abbas Jariani[1,2†], Lieselotte Vermeersch[1,2†], Bram Cerulus[1,2], Gemma Perez-Samper[1,2], Karin Voordeckers[1,2], Thomas Van Brussel[3,4], Bernard Thienpont[3,4,5], Diether Lambrechts[3,4], Kevin J Verstrepen[1,2]*

[1]Laboratory for Systems Biology, VIB-KU Leuven Center for Microbiology, Leuven, Belgium; [2]Laboratory of Genetics and Genomics, CMPG, Department M2S, KU Leuven, Leuven, Belgium; [3]Laboratory for Translational Genetics, Department of Human Genetics, KU Leuven, Leuven, Belgium; [4]VIB Center for Cancer Biology, VIB, Leuven, Belgium; [5]Laboratory for Functional Epigenetics, Department of Genetics, KU Leuven, Leuven, Belgium

**Abstract** Current methods for single-cell RNA sequencing (scRNA-seq) of yeast cells do not match the throughput and relative simplicity of the state-of-the-art techniques that are available for mammalian cells. In this study, we report how 10x Genomics' droplet-based single-cell RNA sequencing technology can be modified to allow analysis of yeast cells. The protocol, which is based on in-droplet spheroplasting of the cells, yields an order-of-magnitude higher throughput in comparison to existing methods. After extensive validation of the method, we demonstrate its use by studying the dynamics of the response of isogenic yeast populations to a shift in carbon source, revealing the heterogeneity and underlying molecular processes during this shift. The method we describe opens new avenues for studies focusing on yeast cells, as well as other cells with a degradable cell wall.

*For correspondence:
kevin.verstrepen@kuleuven.vib.be

[†]These authors contributed equally to this work

## Introduction

Microbes have traditionally been studied on a population level. For example, growth is often measured by the total increase in biomass, and gene expression is analyzed by isolating mRNA from thousands or millions of cells. Even though these methods yield a valuable view on the average behavior of a population, they fail to reveal how each individual microbe contributes to the whole. Therefore, methods to study heterogeneity within a microbial population have been gathering increased interest. Although measuring fluctuations in the total transcriptome of individual cells through scRNA-seq could prove to be of great value in these microbial studies, practical hurdles have so far limited its application in microbiology.

Most of the current methods to study heterogeneity in microbial populations have used a combination of automated microscopy, fluorescent reporter-fusions, specific mRNA species using in-situ hybridization and fluorometric assays. Together, the use of these techniques revealed that individual, genetically identical microbial cells can show different behaviors, including differences in growth speed, gene expression and metabolism (*Campbell et al., 2016*; *Cerulus et al., 2016*; *Cerulus et al., 2018*; *Helaine and Holden, 2013*; *Kiviet et al., 2014*; *Levy et al., 2012*; *Munsky et al., 2012*; *New et al., 2014*; *Papagiannakis et al., 2017*; *Raser and O'Shea, 2004*; *Schwabe and Bruggeman, 2014*; *Takhaveev and Heinemann, 2018*; *Trcek et al., 2012*).

Variation in gene expression is often at the base of different single-cell behaviors. Single-cell mRNA quantification methods were first introduced in the early nineties (*Brady et al., 1990*;

*Eberwine et al., 1992*). More evolved high-throughput methods, in particular scRNA-seq, are now common practice in mammalian cell studies (*Chen et al., 2019*; *Kolodziejczyk et al., 2015*; *Potter, 2018*). However, practical hurdles have restricted their use in microbiology. For example, the microbial cell wall makes implementing current methods more difficult and labor-intensive because of the need to lyse individual cells (*Khan and Yadav, 2004*). Furthermore, the amount of mRNA in comparison to mammalian cells is much lower, with the total number of mRNA molecules in yeast cells reported to be between 20,000 and 60,000, compared to 50,000 to 300,000 mRNA molecules in a typical mammalian cell (*von der Haar, 2008*; *Marinov et al., 2014*; *Pelechano et al., 2010*; *Zenklusen et al., 2008*).

The few existing scRNA-seq studies have employed different methods, most of which being labor-intensive and costly. One important hurdle in many of the existing RNA-seq studies in microbes is the need to isolate single cells prior to further analysis. Traditionally, single cells are isolated one-by-one using manual or laser-based micromanipulation, or through fluorescence-activated cell sorting (FACS). Furthermore, it is reported that the major bottleneck for detection capacity of scRNA-seq methods is located at the initial reverse transcription step where only a fraction of existing single mRNA molecules is captured (*Hwang et al., 2018*; *Islam et al., 2014*). For example, in the first single-cell transcriptome study of eukaryotic microbial cells, five hyphae of *Aspergillus niger* were analyzed, and transcripts were only detected for 4% to 7% of the genes (*de Bekker et al., 2011*). Similarly, two single-cell gene expression studies on bacteria analyzed between four and six individual cells (*Kang et al., 2011*; *Wang et al., 2015*). A scRNA-seq study of malaria parasites from the *Plasmodium* genus reported analysis of 500 individual parasites with transcripts detected for about one third of the total number of genes (*Reid et al., 2018*). Similarly, Saint and coworkers used manual micromanipulation to image and isolate ~2000 single *Schizosaccharomyces pombe* cells and analyzed expression of 18% of the coding genes in these cells (*Saint et al., 2019*). A more recent study that introduced a clever method for strand-specific detection of transcripts in single yeast cells studied 285 single *Saccharomyces cerevisiae* yeast cells grown in rich media, and detected on average 3339 transcripts (*Nadal-Ribelles et al., 2019*).

More recently, microfluidics-based methods have been developed, and these methods generally yield higher throughput while reducing cost and workload. For example, Fluidigm's C1 microfluidic protocol was adapted to study the heterogeneity of *S. cerevisiae* yeast cells in response to osmotic stress (*Gasch et al., 2017*). In this study, 163 cells were analyzed in stressed or unstressed conditions, detecting a population-wide median number of gene transcripts of 2213 in unstressed conditions. Currently, 10x Genomics is arguably the most common commercial microfluidics-based method for scRNA-seq (*Adamson et al., 2016*; *Dixit et al., 2016*; *Kaufmann et al., 2018*; *Yan et al., 2017*; *Zheng et al., 2017*). Here, single cells are trapped in emulsion droplets along with reverse-transcription reagents. Each droplet contains a uniquely labeled primer gel bead, allowing in-droplet barcoding and reverse-transcription of the RNA, before bulk-level sequencing. However, while the method has been optimized for mammalian cells, the protocol cannot be readily applied to yeast cells since the yeast cell wall prevents in-droplet cell lysis.

In this paper, we describe and validate an adaptation to the 10x Genomics protocol that allows using the technology for scRNA-seq in *Saccharomyces cerevisiae*. We apply the method to study transcriptional heterogeneity in yeast populations that are shifted from glucose to maltose. Our previously published results demonstrated that such a shift leads to a strongly heterogeneous response in the cell population, where the time that individual cells require to adapt to the new carbon source (lag time) varies from five to more than 20 hr (*Cerulus et al., 2016*; *Cerulus et al., 2018*; *New et al., 2014*). Moreover, a fraction of cells never resumes growth after the shift to maltose in the observed time window of 24 hr. This pronounced heterogeneity makes the shift from glucose to other carbon sources an excellent example to use scRNA-seq and study if the observed phenotypic heterogeneity is linked to transcriptional heterogeneity.

## Results

### Adapting the 10x genomics scRNA-seq technology for yeast cells

To investigate the observed heterogeneous phenotypic response in isogenic yeast populations when they transition from glucose to maltose on a transcriptional level, we aimed to adapt the 10x

Genomics platform to measure single-cell mRNA concentrations in yeast cells. The 10x Genomics platform is a commercial droplet-based scRNA-seq technology originally developed for mammalian cells (*Vickovic et al., 2016*). Since the yeast cell wall impairs the current lysis step in the protocol, this technology cannot be readily applied to yeast cells. We hypothesized that adding a cell wall digestion enzyme into the reverse transcription master-mix might overcome the current problem and enable the required in-droplet lysis of the yeast cells.

To examine whether the above-mentioned approach could work, we tested whether zymolyase, a cell-wall digestion enzyme, is effective in lysing cells in a temperature regimen similar to that of the 10x Genomics Chromium Single Cell 3′ v2 protocol (*Figure 1A*). Herein, cells are initially stored on ice, subsequently brought to room temperature for droplet generation (~6 min), and droplets are then incubated at 53°C (45 min) to accommodate cell lysis and reverse transcription. For this experiment, the cells were grown in glucose-rich medium (YPD) to exponential phase, and 100 μL of the culture was washed into an ice-cold zymolyase solution. The cells were kept on ice for 25 min, mimicking the time needed in the protocol for preparation of the reagents and microfluidic plate. The cells were then transferred to room temperature for 6 min, and finally to 53°C, mimicking the lysis and reverse transcription steps. Cell counts were monitored throughout the transition steps using an automated cell counter (Bio-Rad TC20). Cell count remained constant during the incubation on ice, and dropped only slightly during the 6 min ambient temperature incubation. After 20 min of incubation at 53°C, cell counts dropped 200-fold, and even below detection limit after 40 min, suggesting that cells were lysing (*Figure 1A*). We confirmed that the heat shock alone will not result in cell lysis, and that zymolyase is necessary for lysis, by comparing the cell lysis efficacy of zymolyase solution to that of PBS during the transition from 0°C to 53°C (*Figure 1—figure supplement 1A*). Based on these observations, we concluded that the adapted protocol should allow lysing of yeast cells in the micro-droplets, with minimal premature out-of-droplet lysis, suggesting that this adaptation of the protocol may open the 10x Genomics platform to not only yeast cells, but also possibly other cell types that require cell wall lysis.

## Media shift schema for scRNA-seq experiment

We next tested the adapted scRNA-seq method to study transcriptional heterogeneity in yeast cells transitioning in a maltose-glucose-maltose shift (*Figure 1B*). After pre-growth in maltose, the cells were transferred to glucose for 12 hr, and then shifted to maltose again. Samples were taken after 6 hr of glucose growth (*glucose-6h*), after 12 hr of glucose growth, just prior to the shift to maltose (*glucose-12h*), 1 hr after the shift to maltose (*lag-1h*), and 3 hr after the shift to maltose (*lag-3h*). Furthermore, a control sample (*mix-glucose-maltose*) consisting of a 50–50% mix of cells actively growing on maltose and cells actively growing on glucose (glucose-12h) was added. Since cells growing on maltose show a distinctly different physiology and gene expression pattern compared to cells growing on glucose, we expected to see two clear sub-groups in the control sample, allowing us to verify whether the protocol works and is able to measure transcription in separate cells. As detailed in the methods section, all samples were processed on the 10x Genomics platform and the generated cDNA libraries were sequenced using the Illumina platform. The number of cells analyzed in the glucose-6h, glucose-12h, lag-1h, lag-3h and mix-glucose-maltose samples was 1695, 1096, 1172, 1469 and 686, respectively (see Materials and methods section for further details), and the per-cell median number of transcripts in each condition was 1261, 894, 528, 607 and 975, respectively. The Seurat package (*Butler et al., 2018*; *Stuart et al., 2019*) was used to filter out cells with excess mitochondrial reads, possible doublets, and to normalize the data.

## scRNA-seq quantification reproduces bulk RNA-seq data

First, we compared data obtained through the single-cell protocol for cells that are exponentially growing on glucose (*glucose-12h*) to bulk RNA-seq data from cultures grown in a similar condition (*Cerulus et al., 2018*). Each unique transcript copy in scRNA-seq data is represented by its unique molecular identifier (*UMI*), which is a 10 bp randomer oligo in the reverse transcription primer. UMI's are widely used in scRNA-seq methods to eliminate quantification inaccuracies due to cDNA amplification (*Islam et al., 2014*). For the purpose of comparing single-cell to bulk data, we summed the UMI count for each gene across the population and log-transformed it after addition of a pseudo-count with a value of 0.1. Expression in bulk RNA-seq was quantified as fragments mapped per kilo-

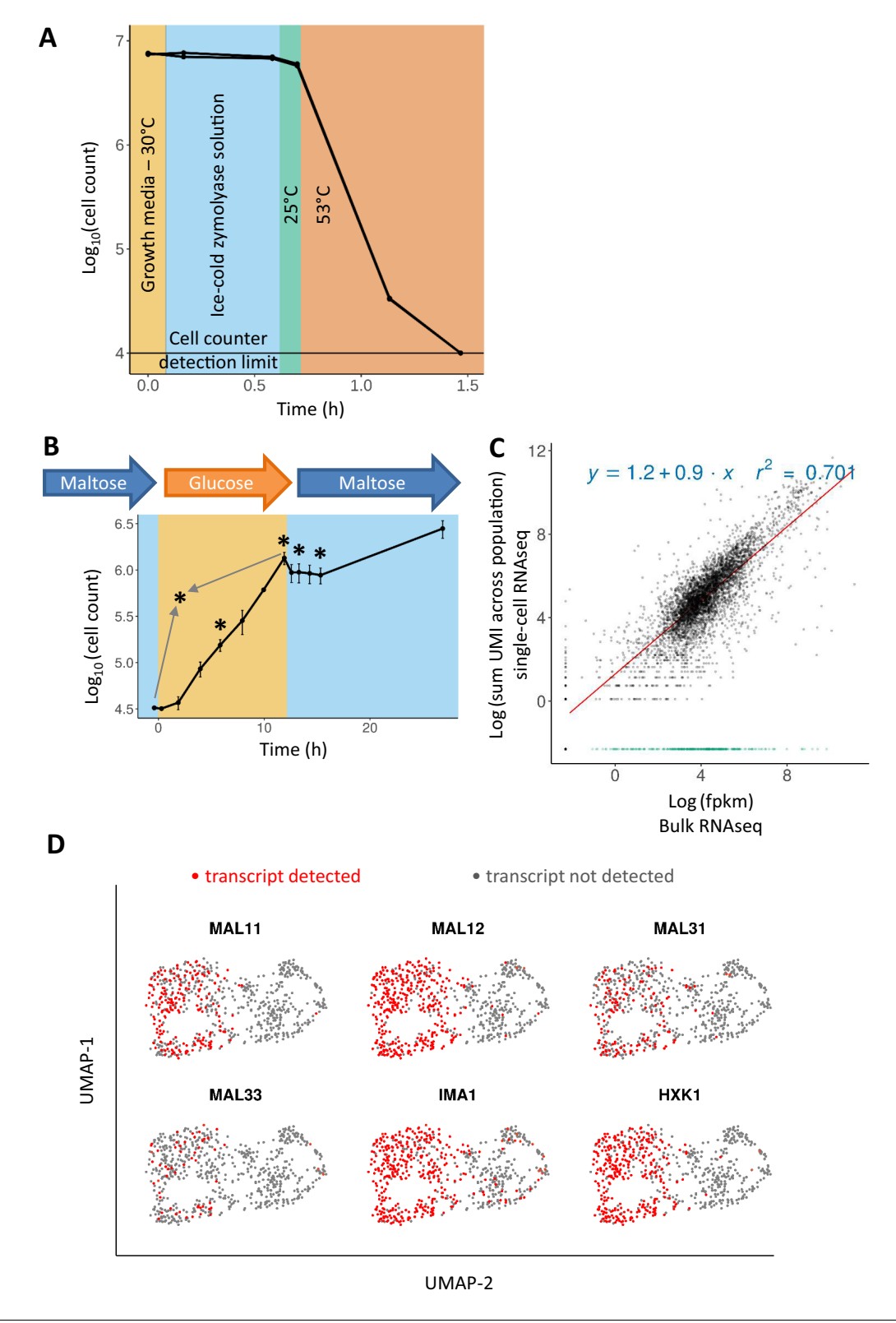

**Figure 1.** Adapting the 10x Genomics' single-cell RNA-seq platform for yeast cells. (**A**) Zymolyase is required and effective for lysing yeast cells in a temperature treatment regimen similar to the 10x Genomics' single cell 3' RNA-seq protocol (see text for details). Cell counts (for biological duplicates) only drop during the incubation at 53℃, showing that cells did not lyse beforehand. (**B**) Sampling scheme and cell counts for the scRNA-seq experiment. Cells were pre-grown in maltose, transferred to glucose for 12 hr, and finally shifted to maltose where they experience a lag phase. After

*Figure 1 continued on next page*

*Figure 1 continued*

the initial maltose pre-growth and subsequent wash to glucose, the cells were diluted for inoculation in glucose. Cell count of these two initial time points is scaled to the dilution level. Sampling points for scRNA-seq are represented by an asterisk: after 6 hr in glucose (referred to as 'glucose-6h'), after 12 hr in glucose ('glucose-12h'), after 1 hr in lag phase ('lag-1h'), after 3 hr in lag ('lag-3h'), and a control sample where cells grown on glucose are mixed in equal parts with cells grown on maltose (grey arrows; 'mix-glucose-maltose'). (C) Correlation between bulk RNA-seq data and scRNA-seq data from cells in similar conditions, but separate experiments. Bulk RNA-seq data from a similar growth condition (12 hr in glucose) were obtained from a previously published experiment (*Cerulus et al., 2018*). The expression levels were quantified as fpkm and log-transformed with addition of a pseudo count. UMI counts from single-cell RNA-seq data were summed across the population, a pseudo-count was added, and log-transformed. The $R^2$ is calculated excluding genes that are detected in bulk RNA-seq and not in scRNA-seq (shown in green). (D) Presence or absence of genes expected to be expressed in maltose distinguishes the two subgroups in the mix of cells grown on glucose and cells grown on maltose.

The online version of this article includes the following figure supplement(s) for figure 1:

**Figure supplement 1.** Adapting the 10x Genomics' single-cell RNA-seq platform for yeast cells.
**Figure supplement 2.** Glucose-growth-specific markers and comparison of expected vs detected number of cells.

base of gene length and per million reads (*fpkm*). The fpkm values were log-transformed as well after addition of a pseudo count. After this transformation of the data, we observe that expression for 42 genes was detected with scRNA-seq but not by bulk RNA-seq (*Figure 1C*), with the majority of these showing a low expression level in the scRNA-seq data. On the other hand, 280 genes were detected only in the bulk RNA-seq. Unlike the first group, expression of these genes reached intermediate to high levels in bulk RNA-seq, while not being detected in scRNA-seq. It is not clear why this set of genes was not detected, as the GC content of these genes is not statistically different from the detected genes (mean %40.6, standard deviation %3.6 vs. mean %40.3, standard deviation %3.5), and they are not enriched for any GO terms. When excluding these 280 genes, the $R^2$ correlation between the single-cell and bulk measurements is 0.7 (*Figure 1C*).

## Heterogeneity detection, sensitivity, and low batch-to-batch variability

Next, we investigated if we could detect the expected two subgroups in the control sample (*mix-glucose-maltose* sample) where we mixed cells grown in glucose with cells grown in maltose prior to mRNA extraction. When the transcriptome data of this control sample is projected into the two-dimensional UMAP (Uniform Manifold Approximation and Projection) space, we indeed observe that the cells are divided into two groups, with only one group showing expression of genes indicative of growth on maltose, such as *MAL11*, *MAL12*, *MAL31*, *MAL32* and *IMA1*, or the glucose- repressed gene *HXK1* (*Figure 1D*).

In order to check for possible technical biases and batch effects, we compared the RNA expression patterns of the culture grown on glucose (glucose-12h) to that of the mix-glucose-maltose control sample. When the data for these two samples are pooled together, we expect half of the cells from the mix-glucose-maltose control sample to be similar to the cells from the glucose-12h sample (*Figure 1—figure supplement 1B*), separated on the transcriptome projection from the maltose-grown fraction in the control sample (*Figure 1—figure supplement 1C*). The analysis shows that this is indeed the case, suggesting that batch effects between samples processed in parallel in the 10x Genomics platform using this protocol are limited, and much smaller compared to the physiological effects.

It is less trivial to find markers specific to the cells growing in glucose, since most of the genes expressed during glucose growth are expected to also be expressed during maltose growth. To screen for possible glucose growth markers, we first marked the cells belonging to maltose growth if they had at least two transcripts of the maltose growth marker genes shown in *Figure 1—figure supplement 1C*. We then carried out differential expression analysis between the cells with maltose growth markers and cells that do not show expression of these specific genes (*Figure 1—figure supplement 1D*).

The barcode vs UMI count plots can be found in *Figure 1—figure supplement 1E*, showing that the number of expected cells is close to, but slightly higher than the number of detected cells. The information from which the number of expected cells in each condition was estimated, is provided in *Table 1* in the Materials and methods section. *Figure 1—figure supplement 1E* shows that a droplet called as 'empty' can still contain 100 UMIs, while the UMI content of a 'filled' droplet is in the range of 1000 to 10,000. Such presence of ambient mRNA in 'empty' droplets might be due to

**Table 1.** Overview of cell numbers during scRNA-seq experiment

| Sample name | Cell count when sampling | Cell count after thaw and wash in PBS | TARGET conc. of cells | Cell count after extra dilution in PBS | Number of cells expected | Number of cells identified after sequencing |
|---|---|---|---|---|---|---|
| glucose-6h | 1.38E+5 | 2.42E+5 | 2.00E+5 | 2.15E+5 | 2150 | 1695 |
| glucose-12h | 1.17E+6 | 4.19E+5 | 2.00E+5 | 2.07E+5 | 2070 | 1096 |
| lag-1h | 8.13E+5 | 3.19E+5 | 2.00E+5 | 1.62E+5 | 1620 | 1172 |
| lag-3h | 7.34E+5 | 3.72E+5 | 2.00E+5 | 2.09E+5 | 2090 | 1469 |
| mix-glucose-maltose (glucose-12h + pre-growth maltose) | 1.91E+6 (pre-growth maltose) | 3.11E+5 | 1.00E+5 (for glucose-12h and pre-growth maltose) | Pre-growth maltose: 6.06E+4 glucose-12h: 1.32E+5 Together: 1.05E+5 | (606+1320)* 0.5 = 963 | 686 |

premature lysis of some cells before they are trapped in the droplets, which agrees with the slight cell count drop observed during ambient temperature incubation of cells with zymolyase (*Figure 1A* –green segment).

## A heterogeneous and yet epigenetically heritable phenotype

After confirming that our method is able to accurately resolve heterogeneous expression, we looked further into the transcriptional heterogeneity in yeast populations throughout a shift from glucose- to maltose-containing medium. When cells are shifted from glucose to maltose, the cells enter growth arrest, the so-called lag phase, during which they adapt to the new carbon source before resuming growth. We previously reported that the duration of the lag phase differs greatly between isogenic cells in a well-mixed, homogeneous population (*Cerulus et al., 2018*). While some cells are able to resume growth after about 5 hr, others take more than 20 hr, and a fraction of cells even completely fails to resume growth at all. Cells can thus be divided into two main groups, one capable of resuming growth (albeit after different lag times), and another group with cells that fail to make the switch. Our previous results showed that respiratory proteins and maltose-catabolic proteins (*MAL* genes) are induced in the fraction of cells that resume growth in maltose (*Cerulus et al., 2018*). However, the molecular source of the heterogeneity remains unclear. For example, no correlation was found between either the replicative age of the cells, nor the level of specific catabolic proteins and the cells' fate in lag phase (*Cerulus et al., 2018*).

## Population transcriptome structure during maltose-glucose-maltose shift

As the lag phase is a transition phase between two steady states of continuous growth on glucose and continuous growth on maltose, dynamics of this transition can be demonstrated using the UMAP projection of the sampled transcriptomes (*Figure 2A* and *Figure 2—figure supplement 1A*). *Figure 2A* shows this projection for the samples glucose-12h, lag-1h, and lag-3h, which are separated in the UMAP space. Similarly, the dynamics of the complete consecutive maltose-glucose-maltose shift is shown in *Figure 2—figure supplement 1A*. As expected, cells growing on maltose (extracted from the mixed sample) are clearly separated from both cells growing on glucose as well as cells in the lag phase. The transcriptome of the cells from the glucose-6h and glucose-12h samples shows some overlap in the UMAP projection, whereas the transcriptome during the lag phase appears to be distinctly different from these continuous growth conditions (*Figure 2A* and *Figure 2—figure supplement 1A*). Interestingly, in the lag-3h sample, the cells appear to be grouped into two sub-populations. Based on cell count measurements (*Figure 1B*), we know that the cells have not yet resumed growth in this condition, hence such presence of population structure might correspond to two distinct groups of cells that eventually will or will not resume growth.

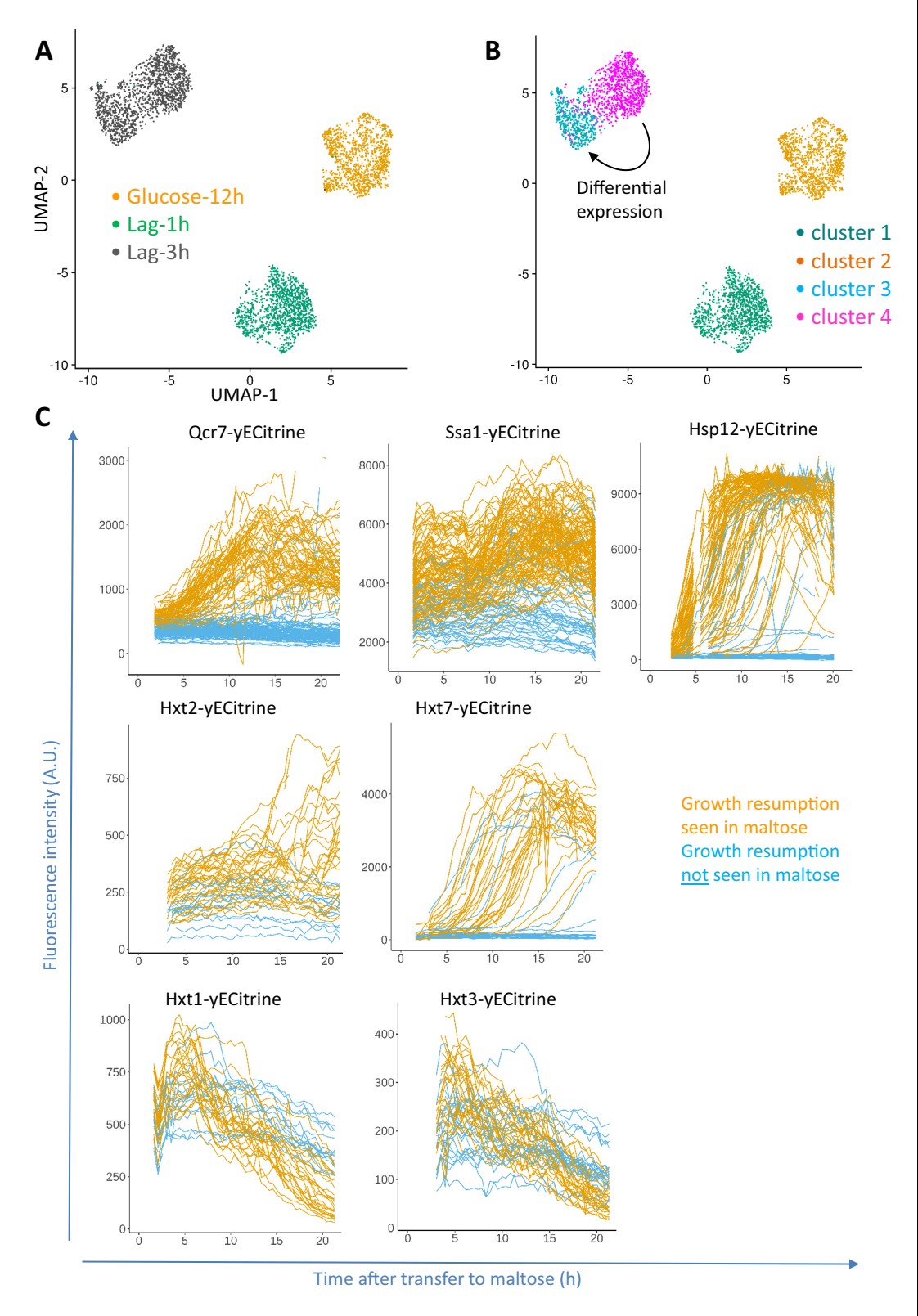

**Figure 2.** Progression of single-cell transcriptomes during the lag phase and validation of the identified differentially expressed genes. (A) Samples for scRNA-seq were collected separately at the end of glucose growth and during the lag phase after the glucose-maltose shift. Expression levels of the samples from glucose-12h, lag-1h and lag-3h were aggregated using the Cell Ranger software. The coloring is based on the sample barcode. (B) Clustering based on expression levels to group the glucose-12h, lag-1h and lag-3h samples into four clusters. The number of clusters was determined

*Figure 2 continued on next page*

*Figure 2 continued*

using the resolution parameter in FindClusters function of Seurat package. The lag-3h sample is grouped into two clusters (clusters 3 and 4). Differential expression analysis between clusters 3 and 4 was performed, of which the list of genes is provided as *Supplementary file 1*. GO enrichment of the overexpressed genes between these two clusters is provided as *Supplementary file 2*. (C) Validation of scRNA-seq results using fluorescent protein fusions. Cells carrying fluorescent fusion reporters were pre-grown in maltose, grown in glucose for 6 to 8 hr, then transferred to maltose and prepared for time-lapse microscopy while in maltose medium. The fluorescence intensity was tracked in each cell over time. The plots for cells that eventually managed to resume growth after transfer to maltose are colored orange, whereas plots for cells that failed to escape the lag phase are plotted in blue. The online version of this article includes the following figure supplement(s) for figure 2:

**Figure supplement 1.** Progression of single-cell transcriptomes during maltose-glucose-maltose shift.
**Figure supplement 2.** Branched expression analysis modeling (BEAM) using Monocle package showing co-expression patterns in continuously growing cultures.
**Figure supplement 3.** Ribosomal gene expression during lag phase.

## Differences in the transcriptome precede phenotypic differences in the lag phase

To further investigate the two subpopulations of the lag-3h sample, we separated the cells from the time series samples of glucose-12h, lag-1h and lag-3h using the *FindClusters* function of the Seurat package. The number of clusters was tuned to four using the *resolution* parameter (*Figure 2B*). The four clusters in *Figure 2B* represent (I) cells from lag-1h sample (cluster 1), (II) cells from glucose-12h sample (cluster 2), and, importantly, two distinct subgroups of cells in the lag-3h sample (clusters 3 and 4).

The genes showing differential expression between the two clusters in the lag-3h sample, that is clusters 3 and 4, were identified using the *FindMarkers* function of the Seurat package. A threshold of 0.05 for false discovery rate and a minimum of 0.25 log2-fold-change difference was used. This analysis identified 628 genes that are overexpressed in cluster four relative to cluster 3; and 374 genes that are overexpressed in cluster three relative to cluster 4 (List provided as *Supplementary file 1*). The corresponding GO terms were extracted from GOrilla (*Eden et al., 2009*) and are provided in *Supplementary file 2*. Cluster 3 is enriched for GO terms such as translation and peptide biosynthetic process. From the 374 genes in this cluster, 116 correspond to ribosomal proteins. It is known that glucose starvation leads to translation halt (*Ashe et al., 2000*), and that translation is downregulated during the lag phase in glucose-maltose shifts (*Cerulus et al., 2018*). Importantly, such higher expression of ribosomal proteins does not necessarily indicate higher protein synthesis in cluster 3, as the cells might not have the energetic requirements for protein synthesis (see further). Instead, it is possible that the elevated expression of ribosomal genes reflects the inability of the cells to react properly to the sudden limitation in glucose availability and shut down protein synthesis. Strikingly, in addition to the ribosomal genes, the low-affinity glucose transporter gene *HXT3* is overexpressed in cluster 3.

Cluster 4 is enriched for GO terms such as oxidation-reduction process, generation of precursor metabolites and energy, energy derivation by oxidation of organic compounds, energy coupled proton transport down electrochemical gradient, and ATP synthesis coupled proton transport. Overexpression of several genes of the electron transport chain complexes agrees with previous observations that activation of respiration is a crucial early step in escaping the lag phase (*Cerulus et al., 2018*; *Perez-Samper et al., 2018*) and suggests that the sub-population of cells that shows activation of these genes may eventually escape the lag phase and resume growth on maltose. Beside these respiratory genes, several stress response genes such as *SSA1* (adjusted p-value<$10^{-50}$) and *HSP12* (adjusted p-value<$10^{-238}$), alongside high-affinity hexose transporters *HXT2* (p-value<$10^{-138}$) and *HXT7* (p-value<$10^{-240}$) are among the overexpressed genes in cluster 4 relative to cluster 3.

These two clusters comprise genes with opposite induction trends during lag phase. Based on this, we hypothesized that clusters 3 and 4 in *Figure 2B* correspond to non-growing and growing cells in lag phase, respectively. Moreover, the cells in cluster four show the transcriptional response that would be expected of cells shortly after a switch from glucose to an alternative sugar, such as the induction of genes responsible for ATP production, suggesting that only cells that are able to generate ATP can eventually escape the lag phase (see also further). Pseudo-temporal ordering of the cells in the process of lag phase yields a similar conclusion with regard to the dynamics of gene

induction in lag phase, as does analyzing the branch dependent expression of the genes (*Figure 2— figure supplement 1B and C*, *Supplementary file 3*). The *branched expression analysis modeling* functionality of the Monocle package was used to find genes that are regulated in a branch-dependent manner. The two branches here correspond to the cells that will or will not resume growth in lag phase. Cluster eight in this figure shows activation of respiratory genes in one of the two branches, and interestingly clusters 2, 3, and 4, which almost exclusively contain ribosomal genes, show stronger deactivation in the branch that activates respiratory genes. This apparent decreasing level of ribosomal genes is not an artifact of normalization due to changes in total transcript levels, since the total UMI counts of these genes actually decreases during lag phase (*Figure 2—figure supplement 1D*).

## Single-cell protein measurements confirm scRNA-seq findings

To test whether the identified differentially expressed genes in the two sub-populations of the lag-3h sample (cluster 3 and 4 in *Figure 2B*, *Supplementary file 1*), are indicative of whether or not cells will resume growth in lag phase, we analyzed the expression of a subset of these proteins by fusing the respective genes with a sequence encoding the fluorescent reporter yECitrine (*Figure 2C*). Specifically, we analyzed a subset of proteins whose expression is upregulated in cluster four relative to cluster 3 (*Figure 2B*): Qcr7, a subunit of complex III of the electron transport chain, Ssa1 and Hsp12, two stress response proteins, and two high-affinity hexose transporters (Hxt2 and Hxt7). In addition, we tested two low-affinity hexose transporters, Hxt1 and Hxt3 (*Yin et al., 2003*). *HXT3* is upregulated in cluster 3, while *HXT1* is not differentially expressed.

Single-cell time-lapse microscopy confirmed that *QCR7*, *SSA1* and *HSP12* indeed show differential expression during the lag phase (*Figure 2C*). In general, cells showing higher expression during the first hours of the lag phase further increase the expression of these genes and eventually escape the lag phase, whereas cells showing lower initial expression are mostly unable to resume growth (*Figure 2C*). As *QCR7* is linked to respiration, the increased activation of this gene confirms previous findings that activation of respiration is vital for efficient escape from the lag phase (*Cerulus et al., 2018*; *Perez-Samper et al., 2018*). The role of higher expression of *HSP12* and *SSA1* is less clear. However, both genes are known to be repressed by glucose, and their induction may simply reflect relief of glucose repression in cells that eventually escape the lag phase. Similarly, the induction of *HXT2* and *HXT7*, encoding high-affinity hexose transporters, is also linked to relief of glucose repression and might serve to help cells take up trace amounts of glucose from the medium.

Together, these data show that cells that eventually escape the lag phase and resume growth on maltose are relieved from glucose repression within about 3 hr of the lag phase. By contrast, in cells that do not escape the lag phase, glucose-repressed genes, including genes related to respiration, stress response, high-affinity glucose transport and metabolism of alternative carbohydrates, remain repressed even after glucose was replaced by maltose.

## Groups of co-expressed genes during glucose growth

Next, we investigated if we could observe expression heterogeneity patterns in the population even prior to the shift to maltose (i.e. the glucose-12h sample). Groups of co-expressed genes in the glucose-12h sample were identified by correlating normalized expression levels across all cells within the population (*Figure 3A*). For this analysis, we employed two filtering criteria. Firstly, the genes should have at least 10 transcript copies across the population, and secondly, the expression of the selected genes must be correlated with at least seven other genes with a minimal Pearson coefficient of 0.1, and a Bonferroni adjusted correlation p-value threshold of 0.05. The groups of co-correlated genes were clustered together using the k-means clustering in Complex Heatmaps R package (*Gu et al., 2016*). The number of clusters was set to six based on visual inspection of the clusters (*Figure 3A*) The cluster assignment for the co-expressed genes in the glucose-12h sample is listed in *Supplementary file 4*. An interactive version of the co-expression heat map of *Figure 3A* can be found at https://abbasjariani.shinyapps.io/sc_co_expression_glucose_12h/.

Ribosomal protein-encoding genes appear to be tightly co-expressed in the glucose-12h sample and are grouped together in cluster 6, which agrees with previous findings (*Gasch et al., 2017*). This cluster contains 94 genes, 87 of which encode ribosomal proteins. Cluster 2 features 116 co-expressed genes including the eight core histone genes, and is enriched for the GO term mitotic cell

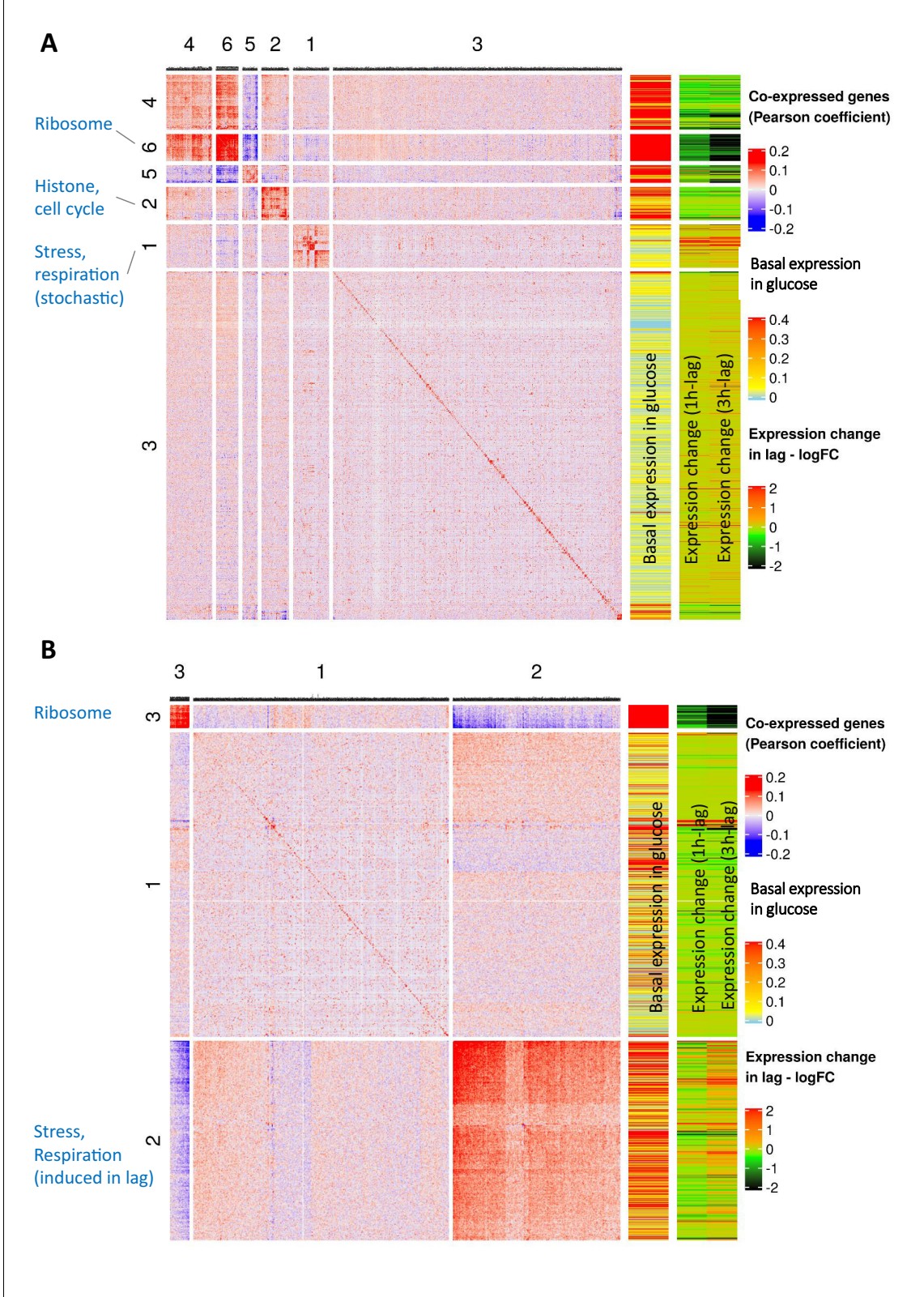

**Figure 3.** Concurrent analysis of co-expression and expression dynamics during glucose growth (glucose-12h) and lag phase (lag-3h). (**A**) Co-expression analysis during glucose growth (glucose-12h) and expression dynamics of genes during glucose-to-maltose shift. This figure is composed of three parts. First, a heat map of expression correlation of the genes (blue-red heat map on the left). Second, a single-column heat map of basal expression level during glucose growth (sky blue-yellow-red heat map in the middle), showing log10 of mean expression level per cell in the population. Third, a two-

*Figure 3 continued on next page*

*Figure 3 continued*

column heat map of expression change during lag phase (black-green-red on the right) which shows log2 fold change of expression in lag-1h and lag-3h samples relative to glucose-12h sample. The color intensity in the expression correlation heat map (left) shows Pearson correlation coefficient of the expression level of gene pairs in the glucose-12h sample. Genes with less than a total of 10 UMI's across all cells were excluded from the analysis. Expression levels were normalized by NormalizeData function of Seurat package. For each gene pair, Pearson correlation coefficient between the two vectors of expression levels across the cells was calculated. Genes that were not correlated with at least seven other genes with a minimal absolute correlation coefficient of at least 0.1 and Bonferroni adjusted p-value threshold of 0.05, were filtered out. The identity diagonal elements were set to zero. Genes were grouped together using hierarchical clustering function of ComplexHeatmaps package. The lists of genes in each of the clusters are provided as *Supplementary file 4*. (B) Co-expression analysis during lag phase (lag-3h) and expression dynamic of genes during glucose-to-maltose shift. Similar to A), but for the lag-3h sample instead of glucose-12h. The lists of genes in each of the clusters are provided as *Supplementary file 4*. The online version of this article includes the following figure supplement(s) for figure 3:

**Figure supplement 1.** Concurrent analysis of co-expression and expression dynamics during lag phase (lag-1h).

cycle process (adjusted p-value<$10^{-9}$). Cluster 5 shows another group of tightly co-expressed genes, the expression of which is inversely correlated with expression of the ribosomal genes in cluster 6 (*Figure 3A*). Similar to cluster 6 (ribosomal genes) and cluster 2 (histone genes), cluster 5 has a high basal expression level in glucose (*Figure 3A*, third column from right). The observation that clusters 6 (ribosomal genes), 2 (histone genes) and 5 all have high basal expression levels, and yet expression of cluster 5 is inversely correlated with the genes in cluster 6, might appear counter-intuitive at first glance. However, considering the seemingly necessary nature of these genes, such inverse correlation might be due to the expression of these genes at mutually exclusive stages of cell-cycle. Cluster 5 consists of 63 members and includes several stress related genes such as *SSA1*, *SSA2*, *SSB1*, *SSB2*, *SSE1 and HSC82*, ergosterol biosynthesis genes *ERG4*, *ERG5* and *ERG6*, and iron metabolism genes *FET3*, *FIT1*, *FIT2 and FIT3*. Cluster 5 also includes multiple genes from the glycolytic pathway (*TDH2*, *TDH3*, *CDC19*, *ENO1*, *ADH1*, *PGI1* and *FBA1*).

Unlike clusters 6, 2 and 5, cluster 1 shows a low basal expression level in glucose (*Figure 3A*). It contains 151 genes, including stress and heat shock genes such as *SSA3*, *SSA4*, *HSP104*, *HSP12*, *HSP26*, *HSP42*, *HSP78*, and *HSP82*. This cluster also includes *CIT1*, a key gene from the TCA cycle, and the high-affinity hexose transporter *HXT7*. We retrieved the transcription factors regulating genes in cluster 1 from Yeastract (*Costanzo et al., 2014*) using default settings. Interestingly, either the Msn2 or Msn4 transcription factors regulate 91 out of 151 members of this cluster. Both transcription factors are shown to play a key role in stochastic gene expression via stochastic nuclear localization, and play a role in survival during environmental shifts (*Hao and O'Shea, 2012*; *Huh et al., 2003*; *Raser and O'Shea, 2004*). This suggests that the members of cluster 1 show a variable and stochastic expression pattern. Cells that express these genes might have an advantage in certain environmental challenges, possibly including a transition to another carbon source, as was the case in our experiment. These data agree with our previous results showing how activation of the TCA cycle and respiration is vital for an efficient transition from glucose to other carbon sources (*Cerulus et al., 2018*; *Perez-Samper et al., 2018*).

## A group of genes stochastically co-expressed during glucose growth, are co-induced in cells resuming growth after a carbon source shift

We carried out a similar co-expression analysis for the samples lag-3h (*Figure 3B*), and lag-1h (*Figure 3—figure supplement 1A*). The co-expression cluster assignment for the genes in these samples is listed in *Supplementary file 4*. Interactive versions of the co-expression heat maps of *Figure 3B* and *Figure 3—figure supplement 1* are provided at: https://abbasjariani.shinyapps.io/inlag3h/ https://abbasjariani.shinyapps.io/inlag1h/.

In the lag-3h and lag-1h samples, similar to the glucose-12h sample, expression of ribosomal genes is highly correlated (cluster three in *Figure 3B* and cluster three in *Figure 3—figure supplement 1A*). However, these genes are not induced during lag phase; instead, their transcript count appears to be decreasing during the lag (right-most column in *Figure 3A,B*; *Figure 3—figure supplement 1A*). Such absence of ribosomal gene transcription agrees with the growth halt during lag phase. Considered together with the previously discussed cell cycle-dependent expression bursts of these genes (*Figure 3A*), it appears that the correlation between expression levels of these genes reflects the cell-cycle stage at which the cells entered the lag phase.

A closer look at the clusters of co-expressed genes in 3h-lag phase sample (*Figure 3B*) reveals that cluster 2 contains genes that are induced after the glucose-to-maltose shift. This might suggest that induction and co-expression of these genes occurs in the cells that resume growth in lag phase. Therefore, we compared these genes to the previously identified overexpressed genes in the cells that will resume growth in the lag-3h sample (cluster four in *Figure 2B*). This comparison reveals that 481 out of the 627 genes that are overexpressed in the subpopulation that will resume growth (cluster four in *Figure 2B*), are among the 906 genes in cluster 2 of co-expressed genes in the lag-3h sample (*Figure 3B*). Moreover, 63 out of the 151 genes in cluster 1 of stochastically (co-)expressed genes in the glucose-12h sample (cluster one in *Figure 3A*) are also present among the 627 genes overexpressed in the cells that resume growth (cluster four in *Figure 2B*). These observations suggest that a group of genes that are repressed during glucose growth, but show stochastic leaky expression, are relieved from repression in a subpopulation of cells that resumes growth after the glucose-to-maltose shift. Furthermore, in line with previous results (*Cerulus et al., 2018*; *Perez-Samper et al., 2018*), these genes play a role in processes such as stress response or respiration.

## Genealogically related cells show more similar behavior

The previous observations suggest that cells growing in glucose show heterogeneous expression of certain genes, and that these differences may affect their ability to resume growth when the carbon source is switched. If this is the case, genealogically related cells, for example a daughter cell that recently budded off a mother cell, might show a similar lag duration, since such cells share a similar chromatin state or a similar concentration of a certain protein or other stochastically fluctuating factor. To test whether the fate of cells upon transfer to maltose is purely stochastic or not, we compared the lag times of genealogical lineages of cells in the population (i.e. different groups of cells that are linked to the same single mother cell) (*Figure 4A,B,C*). Cells were grown in glucose for 8 hr, after which single cells were trapped in a CellAsic microfluidics chamber and were grown in glucose medium inside the chamber for 1 more hour before the medium was shifted to maltose. During the growth on glucose, the trapped single cells divide and form micro-colonies of two to four cells (i.e. one to two cell divisions). If the outcome of the shift is purely stochastic, the lag time of closely related cells within the same micro-colony should not be more similar to each other compared to the rest of the cells. However, we see that the difference in lag times between closely related cells is significantly smaller (Wilcoxon test p-value$<8.68*10^{-13}$) than the mean pair-wise difference across the population (*Figure 4C*). This suggests that genealogically related cells share factors that are stable for at least a few hours, and that the fate of the cell after transfer to maltose is affected by these factors.

To exclude that this observation might be an artifact of uneven flow of media in the microfluidics chamber differentially affecting different micro-colonies, we measured the intracellular signal from a membrane permeable cationic fluorescent dye (JC10, Sigma-Aldrich MAK159) that is flown in combination with maltose media (*Figure 4—figure supplement 1*). An automated image analysis pipeline based on machine learning was developed to measure the intracellular fluorescent signal in single cells and subtract the background signal from the cell periphery (*Figure 4—figure supplement 1A*). We observe that the signal inside the cells increases quickly and homogeneously as soon as the medium containing the dye starts flowing, with all micro-colonies showing a similar response, suggesting a well-distributed flow of medium that reaches all cells (*Figure 4—figure supplement 1B*). Thus, the observed differences between cells in the population are not due to differences in local environments.

## Sustained ATP generation is linked to cell's fate in lag phase

The scRNA-seq results suggest that one important factor that might determine the efficiency with which cells can escape the lag phase is the internal concentration of ATP just prior to the shift. When cells are shifted from glucose to maltose, glucose flux stops and cells can therefore not generate ATP, until they activate maltose import and/or use other ways of generating energy, for example through breakdown of storage carbohydrates. However, it seems plausible that at the moment of the shift, ATP levels could drop below a critical level, leaving the cells unable to adapt to the new conditions, especially if they are unable to restore ATP levels, for example by uptake of trace amounts of glucose and/or activation of the more efficient respiratory metabolism. We therefore

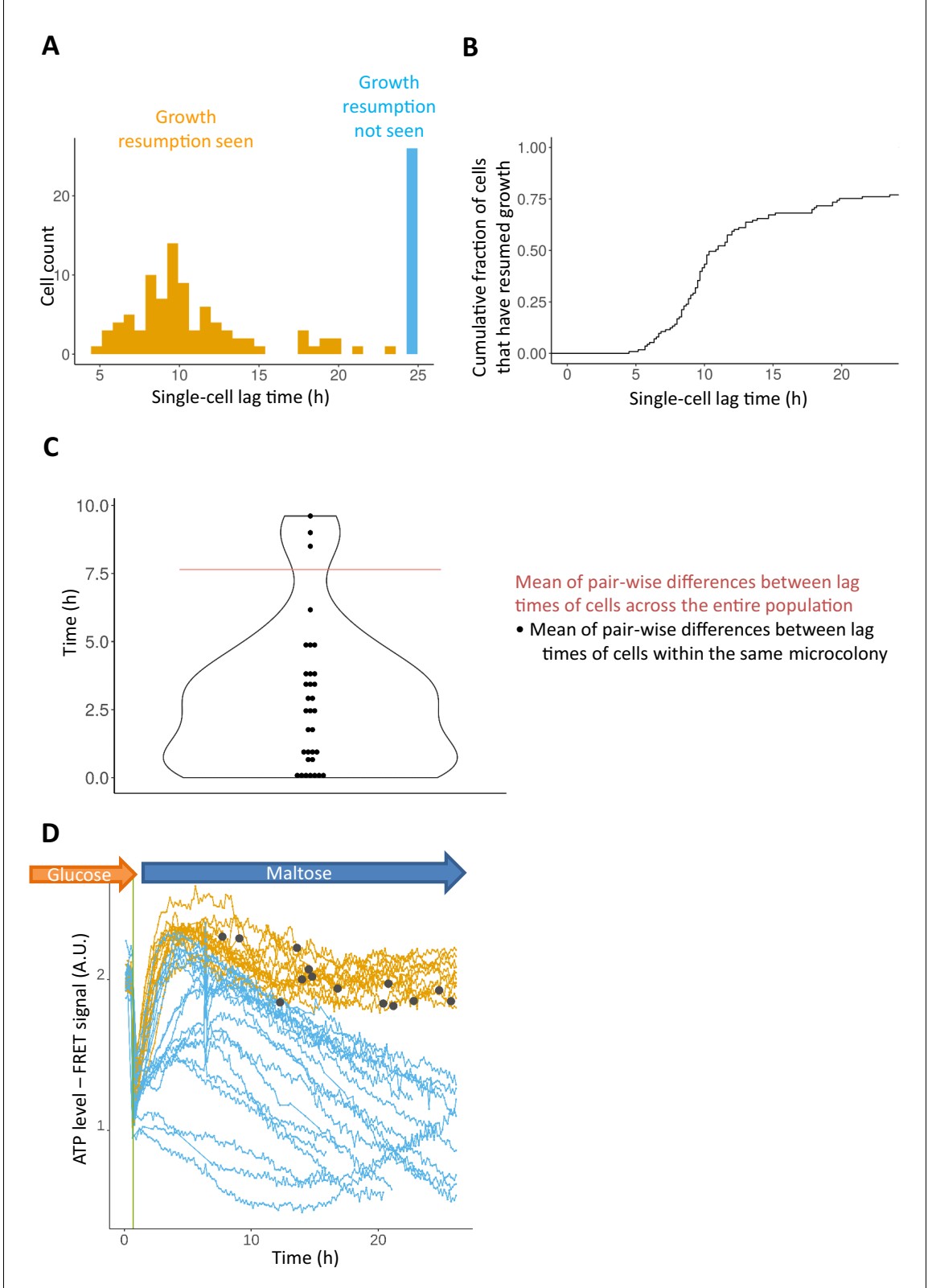

**Figure 4.** Lag phase is heterogeneous, yet not purely stochastic. (**A**) Histogram of single-cell lag times. Colors represent whether the cells resumed growth within the observed time window (orange) or did not resume growth (blue). (**B**) Same data as in A) but shown as a cumulative fraction rather than a histogram (**C**) The lag times of genealogically related cells tend to be more similar to each other compared to the mean pair-wise similarity across the population. Each dot represents the mean pair-wise difference of the lag times within a single micro-colony. The red horizontal line

*Figure 4 continued on next page*

*Figure 4 continued*

represents the mean pair-wise difference of lag times across the population. The median number of cells within each micro-colony is 3. In total 133 single cells were analyzed. (D) Cells expressing a FRET-based ATP sensor (ATeam) were pre-grown in maltose medium, transferred to glucose for 7 hr, and then (at t = 0) loaded into the CellASIC microfluidic chamber with perfusion of glucose-containing medium. After 30 min in glucose the flow was shifted to maltose medium where cells enter the lag phase. Each line represents the FRET signal in a single cell tracked over time. The orange lines represent cells for which growth resumption was seen in 24 hr, while the blue lines represent cells for which growth resumption was not observed. The dark grey circles represent time of growth resumption.

The online version of this article includes the following figure supplement(s) for figure 4:

**Figure supplement 1.** Image analysis pipeline and validation of the microfluidic setup.

measured the level of ATP in live single-cells during glucose-to-maltose shifts (*Figure 4D*). We used a genomically integrated cassette of *ATeam*, a FRET-based sensor for ATP measurements (*Imamura et al., 2009*; *Papagiannakis et al., 2017*). As hypothesized, we observed a sharp drop in ATP levels shortly after the shift to maltose, which coincides with the growth arrest. Most of the cells restore normal intracellular ATP levels within 6 hr after the shift to maltose. Remarkably, only cells that eventually resume growth manage to restore and maintain normal ATP levels during the lag phase, while in cells that fail to resume growth, the ATP level does not recover completely, or quickly drops again after a temporary increase.

Together, these results show that the quick halt in growth after the shift to maltose coincides with a sharp drop in ATP levels and only cells that can regenerate ATP in a sustainable manner are able to resume growth in maltose. Importantly, however, we did not observe differences in internal ATP content between the cells just prior to the shift. This suggests that, while the maintenance and restoration of the ATP content is linked to an efficient escape from the lag phase, it is likely not the primary driver of heterogeneity between cells. Instead, the stochastic differences in the expression of genes linked to glucose transport, stress resistance and/or respiration are likely the true drivers of the observed phenotypic differences.

## Discussion

Given the diversity among individual cells within microbial populations, a high-throughput scRNA-seq method can help to better understand how microbial populations function, both in nature as well as in industrial settings. One of the most widely used methods for scRNA-seq is the 10x Genomics platform. Here, we show how adapting the 10x Genomics protocol to accommodate zymolyase treatment for in-droplet spheroplasting and lysis of cells makes it possible to measure mRNA levels in individual *S. cerevisiae* cells. Our results demonstrate that this novel method corresponds well with previously published studies (*Gasch et al., 2017*; *Jackson et al., 2020*; *Nadal-Ribelles et al., 2019*) in its ability to detect expression heterogeneity. Interestingly, our protocol is similar to another, recently published protocol that also uses the 10x Genomics platform for scRNA-seq in budding yeast, albeit with a separate zymolyase treatment step prior to the droplet generation (*Jackson et al., 2020*). Both methods seem to generate roughly similar output quality, albeit lower than previously reported methods (median number of detected genes per cell are 35–50% of the numbers obtained by Gasch et al. and Nadal-Ribelles et al.). The lower mRNA capture efficiency of these droplet-based methods compared to microwell-based methods and Fluidigm's C1 is consistent with findings in literature (*Ziegenhain et al., 2017*). Hence, the number of analyzed cells versus the mRNA capture efficiency in the different methods seems to be a trade-off. Microwell-based methods typically allow higher mRNA capture efficiencies and thus a better view on the transcriptome at the cost of a lower total number of cells, limiting the analysis of rare phenotypes (such as for example persister cells), and, in theory, an increased risk of undesirable batch effects. Moreover, the ability to increase sample size in droplet-based methods allows the library preparation cost to be greatly reduced to $0.3/cell compared to previously reported costs of $4.15/cell (*Nadal-Ribelles et al., 2019*). Furthermore, our protocol only obtained a sequencing saturation of 70–80% compared to 90% or more for the previously mentioned methods. For our purpose and biological research question, this yielded sufficient information to reveal physiological differences between cells. That said, with deeper sequencing, the mRNA capture efficiency could still be increased. Moreover, it should be noted that the newer chemistry (v3) of the 10x Genomics Chromium Single Cell 3'

Reagent kit has a higher mRNA yield compared to v2 chemistry which is used in our study. Based on the datasets available on the 10x Genomics website which are tested with both chemistries, v3 chemistry yields approximately twice as many UMI's compared to v2 chemistry.

We used the modified 10x Genomics protocol to characterize the dynamics of a yeast population during a shift between carbon sources. The expression correlation analysis of the cells during continuous glucose growth revealed a set of co-expressed genes with low mean expression level that are heterogeneously expressed across the population (*Figure 3A* – cluster 1). This gene set includes genes known to be repressed during exponential growth in glucose, such as several stress response genes, a high-affinity hexose transporter (*HXT7*) and a key respiratory gene (*CIT1*). Functional analysis of these genes might lead to pinpointing the underlying mechanisms of extrinsic noise in expression of these genes. Similar patterns of heterogeneous expression of stress-responsive genes have been attributed to stochastic pulses of nuclear localization of Msn2/4 transcription factors (*Dalal et al., 2014*; *Petrenko et al., 2013*; *Stewart-Ornstein et al., 2013*). We indeed see that 91 out of these 151 stochastically co-expressed genes are regulated by Msn2/4. Interestingly, a high proportion of the genes that show noisy leaky expression in glucose (63 out of 151) are among the genes that show quick induction after transfer from glucose to maltose in cells that will eventually resume growth (*Figure 2B* – cluster 4), suggesting that these stochastically expressed genes might play an important role during lag phase.

Our experiments demonstrate how some cells show stochastic variation in glucose repression, with some cells showing the typical pattern of repression of stress genes and genes involved in respiration, whereas others show somewhat higher transcription of these genes. It is tempting to speculate that these cells might therefore more easily survive in the face of sudden environmental challenges, such as a shift to a less favorable carbon source, since they can quickly induce the necessary genes for survival in the new environment. These new data further support our previous findings that showed how activation of the TCA cycle and respiration is vital for an efficient transition from glucose to other carbon sources (*Cerulus et al., 2018*; *Perez-Samper et al., 2018*). Furthermore , a recent study links specific cellular processes, including hexose transport and mitochondrial function to a bet-hedging strategy whereby some yeast cells in a population grown on glucose tend to prefer fermentation while others opt for respiration (*Bagamery et al., 2020*).

More broadly, our results demonstrate how stochastic variation in gene expression can have important phenotypic consequences, similar to a few other examples where, for example, similar stochastic variation in gene expression determines the phenotypic consequences of mutations (*Burga et al., 2011*; *Casanueva et al., 2012*). As a next step, it would be interesting to further characterize the possible fitness costs and benefits of this stochastic variation in stable and fluctuating environments, and investigate to what extent this variation may in fact be a bet-hedging strategy.

The scRNA-seq method presented here is widely accessible to yeast researchers and is both less labor-intensive and less costly compared to the currently available methods. We used this method to demonstrate the dynamics of adaptation of a microbial population to a new environment. We confirmed that genes that show heterogeneous expression in the scRNA-seq data (*QCR7*, *SSA1* and *HSP12*), indeed show heterogeneous expression in the population, and that this is linked to the ability of cells to subsequently escape the lag phase and resume cell division.

## Materials and methods

### Key resources table

| Reagent type (species) or resource | Designation | Source or reference | Identifiers | Additional information |
|---|---|---|---|---|
| Strain, strain background (*Saccharomyces cerevisiae*) | BY4742 | PMID:9483801 | | S288c MATalpha; *his3Δ1 leu2Δ0 lys2Δ0 ura3Δ0* |
| Strain, strain background (*Saccharomyces cerevisiae*) | KV1156 | PMID:20471265 | | BY4742 *MAL13*:: HygR-*MAL63_c9* |

*Continued on next page*

*Continued*

| Reagent type (species) or resource | Designation | Source or reference | Identifiers | Additional information |
|---|---|---|---|---|
| Strain, strain background (*S. cerevisiae*) | AN62 | PMID:24453942 | | KV1156 *SAL1+* |
| Strain, strain background (*S. cerevisiae*) | AN63 | PMID:24453942 | | AN62 MATa |
| Strain, strain background (*S. cerevisiae*) | AJ78 | This study | | AN63 TEFp-ATeam1.03-KanMX inserted in YRO2 intergenic locus |
| Strain, strain background (*S. cerevisiae*) | BC73 | This study | | AN63 *HXT1*-yECitrine |
| Strain, strain background (*S. cerevisiae*) | BC74 | This study | | AN63 *HXT2*-yECitrine |
| Strain, strain background (*S. cerevisiae*) | BC75 | This study | | AN63 *HXT3*-yECitrine |
| Strain, strain background (*S. cerevisiae*) | BC79 | This study | | AN63 *HXT7*-yECitrine |
| Strain, strain background (*S. cerevisiae*) | MC1 | This study | | AN63 *HSP12*-yECitrine |
| Strain, strain background (*S. cerevisiae*) | MC8 | This study | | AN63 *SSA1*-yECitrine |
| Strain, strain background (*S. cerevisiae*) | MC23 | This study | | AN63 *QCR7*-yECitrine |
| Recombinant DNA reagent | pKT140 | RRID:Addgene_8732 | | KanMX-yECitrine plasmid |
| Recombinant DNA reagent | pTEF:ATP | RRID:Addgene_92179 | | TEF1p-ATeam1.03-KanMX4 plasmid |

## Yeast strains and growth media used

The wild-type strain used in this study is AN63, which is derived from BY4742 (*Brachmann et al., 1998*) by making it maltose-prototrophic (*Brown et al., 2010*), reducing its high-petite frequency (*Dimitrov et al., 2009*) and finally switching its mating type to MATa. In order to track gene expression, protein fusion constructs with the fluorescent marker yECitrine have been constructed for *HXT1*, *HXT2*, *HXT3*, *HXT7*, *HSP12*, *SSA1* and *QCR7*. In order to follow ATP levels, a FRET-based ATP sensor (*Imamura et al., 2009*; *Papagiannakis et al., 2017*) was introduced in the YRO2 intergenic locus. All experiments were performed at 30°C using rich media. The media that were used were YP (10 g/L yeast extract, and 20 g/L peptone) supplemented with 5% glucose or 10% maltose.

## Growth conditions for single-cell lag phase experiments

To measure single-cell lag phases, yeast strains were revived from −80°C onto a YPD plate, or a plate with selective media for strains containing a plasmid. To control cell density of the cultures throughout the experiments, cells were inoculated and serially diluted in 150 µL maltose media (YP-MAL10%), then sealed with a plastic seal and incubated at 30°C, 300 rpm for one day (maximally 24 hr). The next day, the cultures with an $OD_{600}$ ~0.06–0.085 were transferred and again serially diluted to fresh maltose medium and incubated. After pre-growth in maltose, the cells were grown for the desired duration in glucose. For this, cultures with an $OD_{600}$ between 0.065 and 0.085 were washed twice in glucose medium and subsequently serially diluted. The plate was sealed and incubated at

30℃, 300 rpm for the desired duration. Next, these cultures were transferred to the microscope for time-lapse imaging.

## Time-lapse microscopy single-cell lag time measurements

The CellASIC system is a commercial microfluidic system for microscopy, which was used for all time-lapse experiments except for *Figure 2C*. The experiments for *Figure 2C* were performed using the previously published gel-based system for time-lapse microscopy (*Cerulus et al., 2016*; *Cerulus et al., 2018*; *Perez-Samper et al., 2018*) because of its higher throughput. The CellASIC system consists of chambers for trapping the cells, a pump for controlling the media flow through these chambers and software (ONIX perfusion system) for defining the flow program. The chambers can be mounted onto an inverted microscope in order to observe the response of the cells to media shifts (*Zopf and Maheshri, 2013*).

Time-lapse microscopy experiments were carried out according to the CellASIC platform guidelines for yeast (Y04C). When the cultures were ready at the desired growth conditions, we replaced the water in the CellASIC plate with 320 µL of the desired media and loaded 60 µL of cultures with a cell count of ~5*10^6 cells/mL in well 8 (corresponding to an $OD_{600}$ ~0.075–0.085 in an automated plate-reader). Time-lapse pictures were acquired periodically and automatically every 15 min by the Metamorph software (version 7.8.0.0; Molecular Device, LLC), in combination with an inverted Nikon Eclipse Ti microscope equipped with a DL-604M-#VP camera (AndorTM technology), and Lambda XL (Sutter Instruments) light source for fluorescent measurements. The microscope is placed in a temperature-controlled incubator (30℃). An automated image analysis pipeline based on machine learning was developed to quantify lag times and measure the intracellular fluorescent signal in single cells with background signal subtraction from the cell periphery.

## scRNA-seq using the 10x genomics platform

We used the Chromium Single Cell 3' kit (v2) of 10x Genomics for scRNA-seq. We modified the protocol to include zymolyase for digestion of the cell wall. The cells were pre-grown in 3 mL maltose media (YPMAL10%) for two overnights in dilute conditions. The media was refreshed after the first overnight and at each step serial dilutions were made to capture cultures with a cell count of ~2*10^6 cells/mL. At the end of maltose pre-growth, the cells were washed and inoculated into 50 mL glucose media to reach a starting cell density of 10^4 cells/mL. The cells were then grown in glucose for 12 hr, before being washed to maltose media where they experience a lag phase. Cell counts were monitored continuously.

For scRNA-seq, 0.5 mL of culture was mixed with 0.5 mL of glycerol 50% and frozen at −80℃ immediately to store until running the 10x Genomics protocol. For the zymolyase solution, a 100x stock solution (100 mg/mL) was prepared in the buffer of the QuantiTect reverse transcriptase, filter-sterilized (0.2 µm pores), and kept on ice until use to avoid freeze-thaw cycles. The cell cultures were subsequently thawed on ice and resuspended into ice-cold PBS to reach the desired cell count, according to the 10x Genomics Chromium Single Cell 3' protocol. The same steps were followed as detailed in the protocol, with the exception of the reverse-transcription master-mix, where 1 µL of water was replaced with 1 µL of 100x zymolyase solution. The prepared libraries were sequenced on one run of Illumina NextSeq and two lanes of Illumina HiSeq 3000. The 10x Genomics' Cell Ranger pipeline with *mkfastq* option was used to convert the BCL files generated by the Illumina sequencer into fastq files. Cell Ranger with *count* option was used to map and filter the reads and carry out UMI counting. For each sample, the two fastq files from the reads generated by NextSeq and HiSeq sequencers were introduced as arguments to the *count* command. The *expected number of cells* option for the count command was set to the expected value, according to *Table 1*. 17.4 µL of cells were loaded. This volume and the cell counts were chosen based on a target of 1000 cells for the mix-glucose-maltose condition and 2000 cells for the rest of the samples, according to the manual of single-cell 3' kit. The reference genome of S288c was modified to include MAL63, a functional MAL activator, which we introduced in the used strain. In the analyses where samples were aggregated, the *aggr* option of Cell Ranger was used without any aggregation normalization of total UMI's per cell. We did not use normalization of total UMI's since we know from previous experiments that the RNA content of the cells drops during lag phase (*Cerulus et al., 2018*).

The thresholds for filtering out cells with excess mitochondrial reads and possible doublets were determined by inspecting the violin plots of distribution values, using *VlnPlot* function of the Seurat package. The threshold for the maximum total number of genes detected was set to 2000 for glucose-6h, glucose-12h and mix-glucose-maltose samples, and set to 1500 for lag-1h and lag-3h samples. The threshold for the maximum percentage of mitochondrial reads was set to 0.2 for glucose-6h, glucose-12h and mix-glucose-maltose samples, set to 0.5 for the lag-1h sample, and set to 1.5 for the lag-3h sample.

The sequencing saturations for glucose-6h, glucose-12h, lag1h, lag3h, and mix-glucose-maltose samples are respectively 71%, 75%, 85%, 89%, and 82%.

## Acknowledgements

We thank the VIB Nucleomics Core (http://www.nucleomics.be/) for sequencing of the cDNA libraries. We thank VIB Tech Watch for their insights, and providing technology access to 10x Genomics platform. This work was supported by a FWO PhD grant for BC and LV, and a FWO postdoctoral grant for KV. Research in the laboratory of KJV is supported by VIB, AB-InBev-Baillet Latour Fund, FWO, VLAIO, and European Research Council (ERC) Consolidator Grant CoG682009. KJV acknowledges funding from the Human Frontier Science Program (HFSP) grant 246 RGP0050/2013.

## Additional information

### Competing interests

Kevin J Verstrepen: Reviewing editor, *eLife*. The other authors declare that no competing interests exist.

### Funding

| Funder | Grant reference number | Author |
| --- | --- | --- |
| Fonds Wetenschappelijk Onderzoek | | Lieselotte Vermeersch<br>Bram Cerulus<br>Karin Voordeckers |
| Vlaams Instituut voor Biotechnologie | | Kevin J Verstrepen |
| European Research Council | Council CoG682009 | Kevin J Verstrepen |
| AB-InBev-Baillet Latour Fund | | Kevin J Verstrepen |
| Human Frontier Science Program | 246 RGP0050/2013 | Kevin J Verstrepen |

The funders had no role in study design, data collection and interpretation, or the decision to submit the work for publication.

### Author contributions

Abbas Jariani, Conceptualization, Data curation, Software, Formal analysis, Validation, Investigation, Visualization, Methodology, Writing - original draft, Writing - review and editing; Lieselotte Vermeersch, Conceptualization, Data curation, Investigation, Methodology, Writing - original draft, Writing - review and editing; Bram Cerulus, Conceptualization, Validation, Investigation, Methodology, Writing - review and editing; Gemma Perez-Samper, Conceptualization, Investigation, Writing - review and editing; Karin Voordeckers, Conceptualization, Validation, Investigation, Writing - review and editing; Thomas Van Brussel, Resources, Investigation, Methodology; Bernard Thienpont, Resources, Investigation, Methodology, Writing - review and editing; Diether Lambrechts, Resources, Methodology, Writing - review and editing; Kevin J Verstrepen, Conceptualization, Resources, Supervision, Funding acquisition, Methodology, Writing - original draft, Project administration, Writing - review and editing

Author ORCIDs
Abbas Jariani (iD) http://orcid.org/0000-0003-2715-933X
Lieselotte Vermeersch (iD) https://orcid.org/0000-0001-5789-2220
Bernard Thienpont (iD) http://orcid.org/0000-0002-8772-6845
Kevin J Verstrepen (iD) https://orcid.org/0000-0002-3077-6219

Decision letter and Author response
Decision letter https://doi.org/10.7554/eLife.55320.sa1
Author response https://doi.org/10.7554/eLife.55320.sa2

# Additional files

## Supplementary files
• Supplementary file 1. Differential expression analysis between cells assigned to clusters 3 and 4 in UMAP projection of lag-3h sample.

• Supplementary file 2. GO terms corresponding to the differential expression analysis between clusters 3 and 4 in UMAP projection of lag-3h sample.

• Supplementary file 3. Gene cluster assignment, based on Branched expression analysis modeling in pseudo-temporal analysis.

• Supplementary file 4. Cross-correlation matrices of gene expression levels in glucose-12h, lag-1h, and lag-3h samples.

• Transparent reporting form

## Data availability
Sequencing data have been deposited in GEO under accession code GSE144820.

The following dataset was generated:

| Author(s) | Year | Dataset title | Dataset URL | Database and Identifier |
|---|---|---|---|---|
| Jariani A, Vermeersch L | 2020 | Adapting the 10x Genomics Platform for single-cell RNA-seq in yeast reveals the importance of stochastic gene expression during the lag phase | https://www.ncbi.nlm.nih.gov/geo/query/acc.cgi?acc=GSE144820 | NCBI Gene Expression Omnibus, GSE144820 |

The following previously published dataset was used:

| Author(s) | Year | Dataset title | Dataset URL | Database and Identifier |
|---|---|---|---|---|
| Jariani A, Cerulus B | 2018 | Transition between fermentation and respiration determines historydependent behavior in fluctuating carbon sources | https://www.ncbi.nlm.nih.gov/geo/query/acc.cgi?acc=GSE116246 | NCBI Gene Expression Omnibus, GSE116246 |

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
