## [Decision Letter]

**Acceptance summary:**

This study describes a new high-throughput and straightforward-to-implement method for measuring single-cell transcriptomes in budding yeast. Application of this method to a carbon source shift (glucose to maltose) provides support for a model in which growth resumption is correlated with a specific mRNA signature characteristic of a subpopulation of cells early during the response. The study's novelty primarily lies in the development of a novel single cell RNA (scRNA) sequencing method that has potential to provide novel insights into budding yeast biology.

**Decision letter after peer review:**

Thank you for submitting your article "Adapting the 10x Genomics platform for single-cell RNA-seq in yeast reveals stochastic gene expression during lag phase" for consideration by *eLife*. Your article has been reviewed by two peer reviewers, and the evaluation has been overseen by a Reviewing Editor and Naama Barkai as the Senior Editor. The reviewers have opted to remain anonymous.

The reviewers have discussed the reviews with one another and the Reviewing Editor has drafted this decision to help you prepare a revised submission.

As an aside: The agreement was that "10x Genomics" was not necessary for the title. "Single-cell RNA-seq reveals stochastic gene expression during lag phase in budding yeast" would work as well.

Summary:

This manuscript describes a study in the budding yeast *Saccharomyces cerevisiae* addressing the response to a carbon source shift (glucose to maltose). The authors show that the 10x Genomics platform can be adapted for yeast, allowing the measurement of single-cell transcriptomes during this adaptation response. The technical advance of this manuscript is that this method would allow for a considerable scale-up (to 1500 cells) relative to previously 96 well plate methods and a reduction in cost for library preparation per cell (from $4.15 to $1.00, or even $0.30 if scaled up to 5,000 cells), which will open up new areas of research for yeast biologists. Application of this method to a carbon source shift provides support for a model in which growth resumption is correlated with a specific mRNA signature characteristic of a subpopulation of cells early during the response. The paper provides novel insights into an interesting aspect of yeast biology (lag phase) by combining multiple approaches (e.g. microfluids, microscopy) and primarily by developing a single cell RNA (scRNA) sequencing).

Essential revisions:

1) As it is written, the paper is a mixture of a method and a research story. Given the novelty of the scRNA sequencing technique and that the manuscript type is "Tools and resources", the authors should revise the manuscript such that the work is presented from the method angle and that the carbon source shift experiment is an example application of the utility of the novel method. For example:

1a) Seems strange that the method is presented only in the third paragraph of the Results section.

1b) The actual need to scale up beyond the 100 cell level for most yeast studies is unclear and unjustified in the manuscript. The authors should make a clear and persuasive argument on this point.

2) The authors need to address the relatively poor mRNA capture efficiency for their method compared to that in two recent reports (Gasch et al. and Nadal-Ribelles et al.). As they point out, newer chemistry available from 10x Genomics alone might get them closer to previously reported capture efficiencies. At present, the relatively poor mRNA capture efficiency means that this method may not be generally appealing.

3) There is a very recent paper developing a similar scRNA- seq: Jackson et al., 2020, that analysed a much larger number of sc (38K). This is not a problem given *eLife*'s vision, but the authors should discuss how the methods compare.

---

## [Author Response]

As an aside: The agreement was that "10x Genomics" was not necessary for the title. "Single-cell RNA-seq reveals stochastic gene expression during lag phase in budding yeast" would work as well.

We have changed the title according to the reviewers’ suggestion to “A new protocol for single-cell RNA-seq reveals stochastic gene expression during lag phase in budding yeast”.

Essential revisions:1) As it is written, the paper is a mixture of a method and a research story. Given the novelty of the scRNA sequencing technique and that the manuscript type is "Tools and resources", the authors should revise the manuscript such that the work is presented from the method angle and that the carbon source shift experiment is an example application of the utility of the novel method. For example:1a) Seems strange that the method is presented only in the third paragraph of the Results section.

Agreed; We have extensively rewritten and re-arranged the manuscript so that we approach it from the method angle, using the carbon source shift experiment more as an example.

1b) The actual need to scale up beyond the 100 cell level for most yeast studies is unclear and unjustified in the manuscript. The authors should make a clear and persuasive argument on this point.

Scaling up beyond the 100-cell level will not only yield higher statistical power, and lower the risk of undesirable batch effects, it also opens the possibility to study more rare phenotypes in microbial populations.

We have added a few arguments in the Discussion section of the manuscript (first paragraph).

2) The authors need to address the relatively poor mRNA capture efficiency for their method compared to that in two recent reports (Gasch et al. and Nadal-Ribelles et al.). As they point out, newer chemistry available from 10x Genomics alone might get them closer to previously reported capture efficiencies. At present, the relatively poor mRNA capture efficiency means that this method may not be generally appealing.

Linking back to the previous comment (1b), we do think that the trade-off between mRNA capture efficiency and the number of cells analyzed is important to consider. Using methods with higher capture efficiency generally comes at the cost of low number of analyzed cells. Moreover, we only achieved a sequencing saturation of 70-80%, so deeper sequencing could also still increase the capture efficiency; and the newly introduced chemistry for the 10X platform probably also helps.

We have added a brief discussion about these points in the Discussion section (first paragraph).

3) There is a very recent paper developing a similar scRNA- seq: Jackson et al., 2020, that analysed a much larger number of sc (38K). This is not a problem given eLife's vision, but the authors should discuss how the methods compare.

Thanks for bringing this to our attention. The major difference between the two methods comes down to spheroplasting before loading the cells into the 10X device vs. immediate cell lysis in the droplets. We believe that our protocol with in-droplet cell lysis further simplifies the protocol. This recent paper indeed analyzed a much larger number of cells, but the number of single-cells analyzed can be decided upon by the user (calculations via the Cell Suspension Volume Calculator Table in the manual of the 10X system), depending on, among other things, sequencing cost and number of cells needed for biologically relevant conclusions. It is worth noting that the median number of detected genes is quite comparable between the two methods, and this would likely also depend on the specific conditions of the experiment, which determines how many genes are expressed. Furthermore, the much larger number of single-cells mentioned is actually the sum over 11 different conditions, accounting to around 3500 cells per sample, which is only about 1.5-2 times more than what we aimed for in our experiment.

We have included a short discussion on the comparison between the two methods in the manuscript (Discussion, first paragraph).